# Development of a method of passaging and freezing human iPS cell-derived hepatocytes to improve their functions

Jumpei Inui[1], Yukiko Ueyama-Toba[1,2,3], Seiji Mitani[1¤], Hiroyuki Mizuguchi[1,2,3,4,5]*

1 Laboratory of Biochemistry and Molecular Biology, Graduate School of Pharmaceutical Sciences, Osaka University, Osaka, Japan, 2 Laboratory of Functional Organoid for Drug Discovery, National Institute of Biomedical Innovation, Health and Nutrition, Osaka, Japan, 3 Integrated Frontier Research for Medical Science Division, Institute for Open and Transdisciplinary Research Initiatives, Osaka University, Osaka, Japan, 4 Global Center for Medical Engineering and Informatics, Osaka University, Osaka, Japan, 5 Center for Infectious Disease Education and Research (CiDER), Osaka University, Osaka, Japan

¤ Current address: Advanced Medical Science of Thrombosis and Hemostasis, Nara Medical University, Nara, Japan

* mizuguch@phs.osaka-u.ac.jp

**Data Availability Statement:** All relevant data are within the manuscript and its Supporting Information files.

## Abstract

Human induced pluripotent stem (iPS) cell-derived hepatocyte-like cells (HLCs) are expected to replace primary human hepatocytes as a new source of functional hepatocytes in various medical applications. However, the hepatic functions of HLCs are still low and it takes a long time to differentiate them from human iPS cells. Furthermore, HLCs have very low proliferative capacity and are difficult to be passaged due to loss of hepatic functions after reseeding. To overcome these problems, we attempted to develop a technology to dissociate, cryopreserve, and reseed HLCs in this study. By adding epithelial-mesenchymal transition inhibitors and optimizing the cell dissociation time, we have developed a method for passaging HLCs without loss of their functions. After passage, HLCs showed a hepatocyte-like polygonal cell morphology and expressed major hepatocyte marker proteins such as albumin and cytochrome P450 3A4 (CYP3A4). In addition, the HLCs had low-density lipoprotein uptake and glycogen storage capacity. The HLCs also showed higher CYP3A4 activity and increased gene expression levels of major hepatocyte markers after passage compared to before passage. Finally, they maintained their functions even after their cryopreservation and re-culture. By applying this technology, it will be possible to provide ready-to-use availability of cryopreserved HLCs for drug discovery research.

## Introduction

Primary (cryopreserved) human hepatocytes (PHHs) are the main cell source used for preclinical *in vitro* studies of drug metabolism and disposition. However, PHHs have some issues, such as lot-to-lot variability, limited supply, and loss of hepatic functions in culture. Human induced pluripotent stem (iPS) cell-derived hepatocyte-like cells (HLCs) are expected to replace PHHs as a new stable source of functional hepatocytes. Many studies have been

**Funding:** This research was financially supported by Japan Agency for Medical Research and development, AMED (grant number JP21be0304202, JP23fk0310512, JP23mk0101213) and the Japan Society for the Promotion of Science (JSPS) KAKENHI [grant number JP21K18247].

**Competing interests:** The authors have declared that no competing interests exist.

conducted to generate HLCs for drug discovery research. For example, disease models *in vitro* have been established using HLCs differentiated from patient-derived iPS cells [1–3]. Human iPS cells derived from individuals with single nucleotide polymorphisms (SNPs) in cytochrome P450 (CYP) 2D6 recapitulated the poor metabolizer phenotype by differentiation into HLCs [4]. In addition, genome editing technology successfully recapitulated the phenotypes of poor metabolizers in HLCs and revealed the contribution of specific CYP enzymes in pharmacokinetics [5, 6]. Moreover, hepatocyte transplantation technologies using HLCs sheets [7], spheroids [8], and organoids [9] have been developed as an alternative to living donor liver transplantation.

Although HLCs are expected to be utilized for pharmaceutical applications, several issues remain. The first is that HLCs have low drug-metabolizing enzyme activities [10]. They are reported to be similar to fetal hepatocytes rather than adult hepatocytes [11]. Several groups have tried to sort high-functioning HLCs from their low-functioning counterparts using hepatocyte-specific cell surface markers such as asialoglycoprotein receptor 1 (ASGR1) [12] and sodium taurocholate cotransporting polypeptide (NTCP) [13]. However, the utility of the sorted HLCs is limited, because they tend to lose their functions after reseeding. We previously succeeded in expressing a neomycin resistance gene downstream of the CYP3A4 gene in HLCs, which allowed purification of only high-functioning HLCs expressing CYP3A4 by neomycin selection. However, this approach necessitated the use of genome editing to introduce the neomycin resistance gene to human iPS cells [14]. In addition, the span of about a month is required to differentiate HLCs from human iPS cells. Many researchers have developed alternative hepatocyte differentiation protocols from human iPS cells [15–18], but most of them have not been able to shorten the culture period. Finally, HLCs have a very low proliferative capacity and are difficult to be passaged due to their loss of hepatic functions after reseeding. Due to these problems, HLCs have not replaced PHHs in pharmaceutical research. Development of a method to overcome these barriers would facilitate the wider use of HLCs.

In this study, we developed a technology to dissociate, cryopreserve, and reseed HLCs without loss of their functions. We then passaged HLCs by this novel method and evaluated their functions.

## Results

### Effects of passage on the functions of HLCs

First, we examined whether it was indeed impossible to passage and re-culture HLCs. For this purpose, we prepared HLCs according to our previously reported method [4], and then passage-culture of HLCs as shown in **Fig 1A.** Cell morphology of HLCs was observed by phase-contrast microscopy before passage and again 7 days after passage. The gene expression levels of hepatocyte markers (*albumin, ALB; hepatocyte nuclear factor 4 alpha, HNF4α; cytochrome P450 3A4, CYP3A4*) were analyzed by real-time RT-PCR. After passage, HLCs lost their polygonal morphology and showed a fibroblast-like morphology (**Fig 1B**). The gene expression levels of hepatocyte markers were significantly lower in HLCs after passage than in those before passage. The *ALB* expression level in HLCs after passage decreased by about 50-fold compared to that before passage (**Fig 1C**). The gene expression level of CYP3A4 increased in HLCs after passage, while the expression levels of CYP2C9 and CYP2C19 did not change significantly (Figs 1C and S1). These results suggested that HLCs lost their functions after passage.

To identify the cause of the loss of hepatic functions in HLCs following the passage operation, we analyzed the differences in gene expression in HLCs between before and after passage in more detail. The gene expression levels of mesenchymal cell markers (*snail family transcriptional repressor 1, SNAI1; Fibronectin*) increased, while those of epithelial cell marker *(E-*

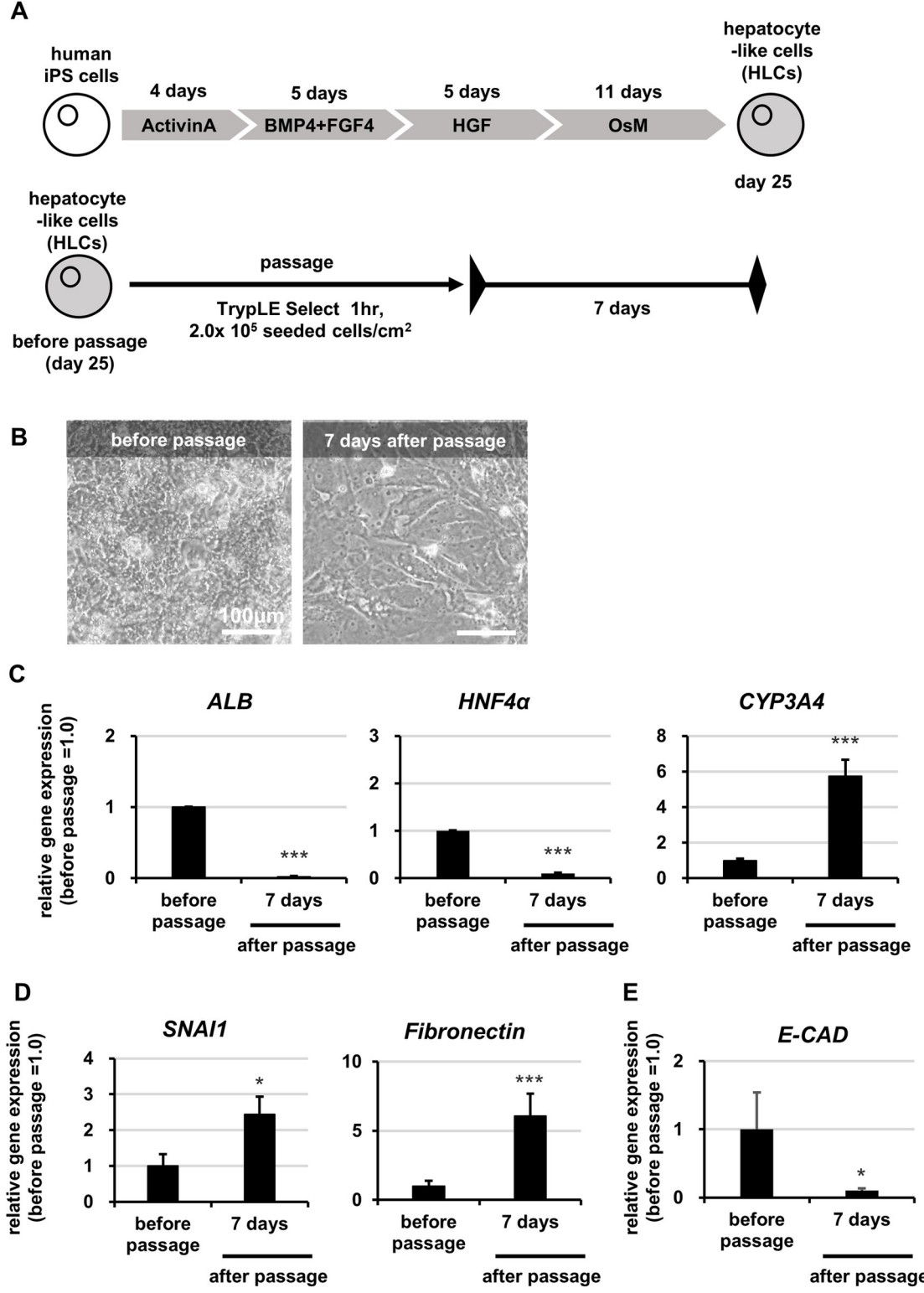

**Fig 1. Effect of passage on HLCs.** Human iPS cells (Tic) were differentiated into hepatocyte-like cells (HLCs) as described in the Materials and Methods section. (A) The schematic overview shows the protocol for hepatic differentiation and passage of HLCs. (B) Phase-contrast micrographs of HLCs before passage and HLCs cultured for 7 days after passage are shown. (C, D) The gene expression levels of hepatocyte (C) (*ALB*, *HNF4α*, *CYP3A4*), mesenchymal (D) (*SNAI1* and *Fibronectin*) or epithelial (E) (*E-CAD*) markers were examined in HLCs before passage and HLCs cultured for 7 days after passage by real-time RT-PCR. The gene expression levels in HLCs before passage were taken as 1.0. Data represent the means ± SD (n = 3). Statistical significance was evaluated by unpaired two-tail Student t test (*$p < 0.05$, **$p < 0.01$, ***$p < 0.005$: compared with "before passage").

*cadherin*, *E-CAD*) decreased in HLCs after passage (**Fig 1D**). Thus, the passage and re-culture might cause epithelial-mesenchymal transition (EMT) in HLCs and thereby eradicate their functions as hepatocytes.

## Effects of EMT inhibitors on the functions of HLCs after passage

We next attempted to improve the hepatic functions in HLCs after passage by inhibiting EMT. We examined the effects of a MEK inhibitor PD0325901 (P), a TGFβ inhibitor SB431542 (S), and a ROCK inhibitor Y-27632 (Y), which have previously been reported to inhibit EMT [19, 20]. Following the protocol shown in **Fig 2A**, the functions in HLCs after passage were examined when these compounds were administered alone or in combination. The results showed that the combined treatment of P, S and Y (PSY group) improved the cell morphology of HLCs after passage (**Fig 2B**) and increased hepatocyte marker gene expression (**Fig 2C**). In addition, the gene expression levels of mesenchymal cell markers (*SNAI1*, *Fibronectin*) decreased, while those of the epithelial cell marker (*E-CAD*) increased in the PSY group compared to the control group (**S2A Fig**). The PSY treatment itself did not greatly increase the gene expression levels of hepatocyte markers in the HLCs without passage (**S2B Fig**). These results suggested that PSY treatment could improve hepatic functions in HLCs after passage by inhibiting EMT. On the other hand, the improvement of cell morphology by PSY treatment was not sufficient, and the gene expression levels of the hepatocyte markers (*ALB*, *HNF4α*) were still low (**Fig 2C**), suggesting that further improvement of the passage manipulation was needed.

## Optimum cell-dissociation time for passage of HLCs

Passage of HLCs required 1 hr of cell dissociation time to recover as many cells as possible. This was a longer dissociation time than typically required for ordinary cell lines. Because there was a risk that the HLCs would be injured by the cell dissociation enzyme, resulting in decreased hepatic functions after passage, we investigated the optimum cell dissociation time for passage. Following the protocol shown in **Fig 3A**, individual HLCs were recovered after four different cell-dissociation times. Then the number of recovered cells and cell viability under each condition were evaluated. As the cell dissociation time shortened, the recovery of HLCs and cell viability decreased, although there was no significant difference in the cell viability (**Fig 3B and 3C**). Next, we passaged the HLCs under each dissociation time and cultured them to evaluate the cell morphology and the gene expression levels of hepatocyte markers. Cell morphology was worse in the 60 min of dissociation (60-min dissociation group), but greatly improved after 15 min of dissociation (15-min dis sociation group). The morphology in the 15-min dissociation group was similar to that before passage (**Fig 3D**). With the shortening of cell dissociation time, the gene expression levels of hepatocyte markers in HLCs after passage increased (**Figs 3E and S3**). This trend was also observed in another human iPS cell line, the DOO line (**S4 Fig**). Note that the experiment could not be performed in the 5-min dissociation group because the number of cells recovered was very small. These results suggested that shortening the cell dissociation time suppressed the decline of hepatic gene expressions, while long duration of the cell dissociation might damage HLCs and decreased hepatic functions. We further examined which cell dissociation enzyme was the most suitable for passage of HLCs. When the TrypLE Select was used, the cells showed the highest gene expression levels of hepatocyte markers and the highest cell recovery (**S5 Fig**). Therefore, we decided to use 15-min dissociation with TrypLE Select for HLCs passage.

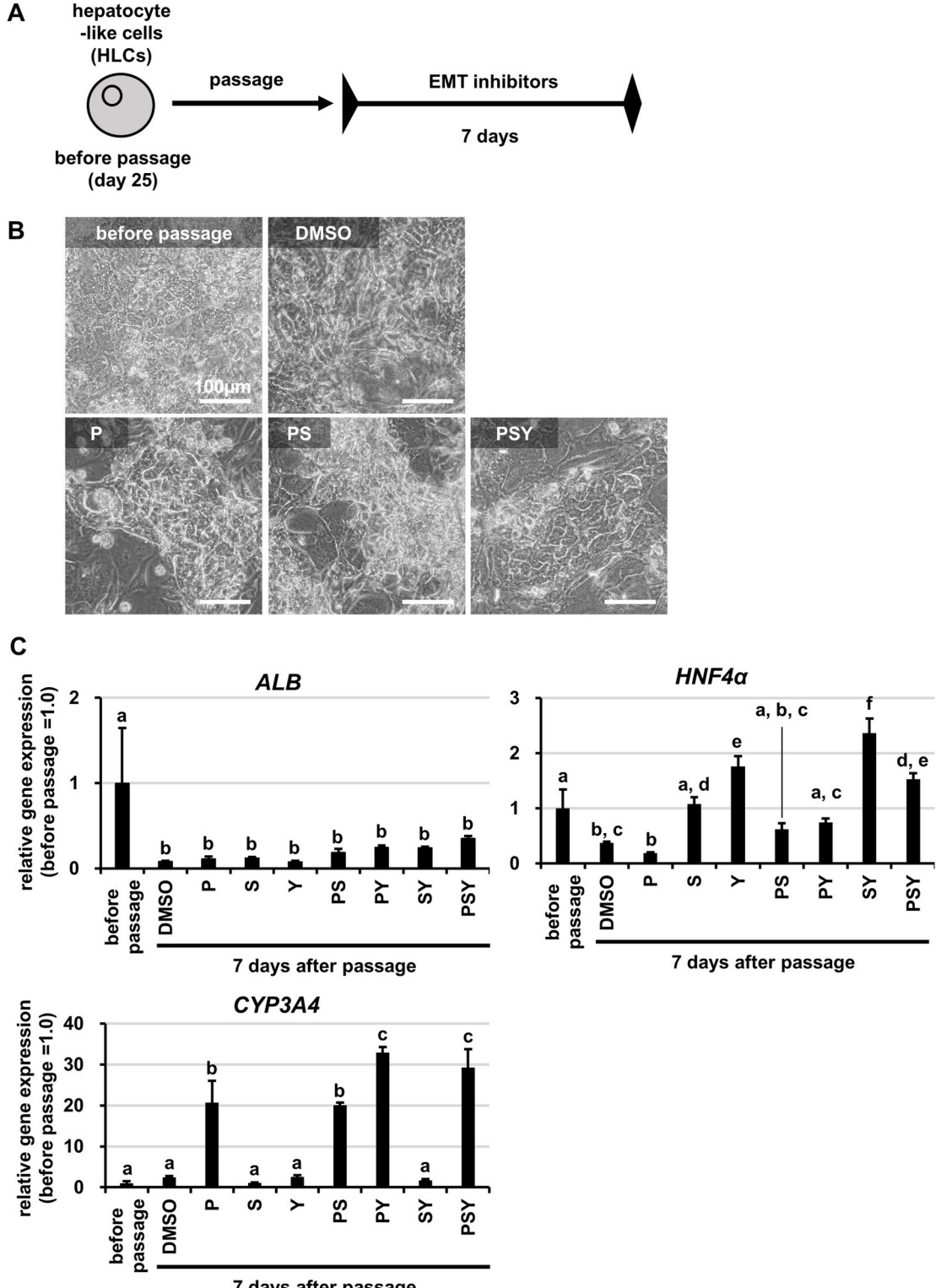

**Fig 2. Effects of small molecule compounds on HLCs after passage.** Human iPS cells (Tic) were differentiated into hepatocyte-like cells (HLCs) as described in the Materials and Methods section. (A) The schematic overview shows the protocol for passage of HLCs. HLCs were passaged and cultured for 7 days with vehicle control (DMSO), with a single EMT inhibitor (P, 0.5 μM PD0325901; S, 2 μM SB43154; Y, 10 μM Y-27632), and a combination of EMT inhibitors (PS, PY, SY, PSY). (B) Phase-contrast micrographs of HLCs before passage and HLCs after passage cultured with each condition are shown. (C) The gene expression

levels of hepatocyte markers (*ALB*, *HNF4α*, *CYP3A4*) were examined in HLCs before passage and HLCs after passage cultured with each condition for 7 days by real-time RT-PCR. The gene expression levels in HLCs before passage were taken as 1.0. Data represent the means ± SD (n = 3). Statistical significance was evaluated by one-way ANOVA followed by Tukey's post-hoc tests to compare all groups. Groups that do not share the same letter are significantly different from each other (p<.05).

### Evaluation of HLCs passaged under the optimized passage conditions

Based on the results shown in Figs 1–3, we have developed a passage method for HLCs (**Fig 4A**). In this protocol, the cell dissociation time was shortened, and PSY were added as EMT inhibitors. Various analyses were performed on HLCs cultured for 7 days after passage under this condition.

The activity of CYP3A4, a major drug-metabolizing enzyme expressed in human hepatocytes, was evaluated in HLCs before and after passage. The results showed that CYP3A4 activity in HLCs after passage was higher than that in PHHs cultured for 48 hr (PHH 48hr) or in HLCs before passage (**Fig 4B**). Therefore, it was suggested that the drug-metabolizing enzyme activity in HLCs was enhanced by use of the appropriate passage method. Immunofluorescence staining confirmed that hepatocyte marker proteins (ALB, HNF4α CYP3A4, and αAT) were expressed in HLCs after passage as well as before passage (**Fig 4C**). FACS analysis was performed to further investigate the levels of ALB and CYP3A4 protein expression in HLCs after passage. The results showed that the percentage of ALB- or CYP3A4-positive cells increased in the HLCs after passage compared to that before passage, suggesting that the HLCs matured into more highly functional hepatocytes by passage (**S6 Fig**). The capacity of low-density lipoprotein (LDL) uptake in HLCs was evaluated using fluorescently labeled LDL. Intracellular uptake of LDL was observed in HLCs after passage, indicating that they maintained LDL uptake capacity after passage (**Fig 4D**). PAS (periodic acid-Schiff) staining was performed to evaluate the glycogen storage capacity of HLC. As before passage, HLCs after passage were stained by PAS, suggesting that they had glycogen storage capacity (**Fig 4E**). We further compared the gene expression levels of major hepatocyte markers in HLCs after passage with those in PHHs, HLCs before passage and those in HepG2 cells (**Fig 4F**). The gene expression levels of most markers in HLCs after passage were comparable or higher than those in HLCs before passage. In particular, the gene expression levels of the *ALB* and CYP genes (*2B6*, *2C9*, *2C19*, *3A4*) in HLCs after passage were higher than those in PHH 48hr. We confirmed that expression levels of many hepatocyte marker genes in HLCs were more increased by passage comparing to extended culture without passage (**S7 Fig**). These results indicated that the passage method developed in the present study not only maintained hepatic functions, but also upregulated hepatic gene and protein expression and CYP3A4 activity in HLCs.

### Extended culture of HLCs after passage

We also examined whether the hepatic functions in HLCs after passage could be maintained for a longer period. Following the protocol shown in **Fig 5A**, we examined the cell morphology and the gene expression levels of hepatocyte markers in HLCs after passage. The results showed that the HLCs after passage maintained their polygonal morphology for 15 days (**Fig 5B**). The expression levels of *ALB*, *HNF4α* and *CYP3A4* peaked at 3 days after passage (**Fig 5C**). Thereafter, their expression levels showed a decreasing trend, but remained at the same or higher levels as compared with those before passage until 15 days after passage (**Fig 5C**). These results suggested that HLCs maintained higher hepatic functions for at least 2 weeks after passage.

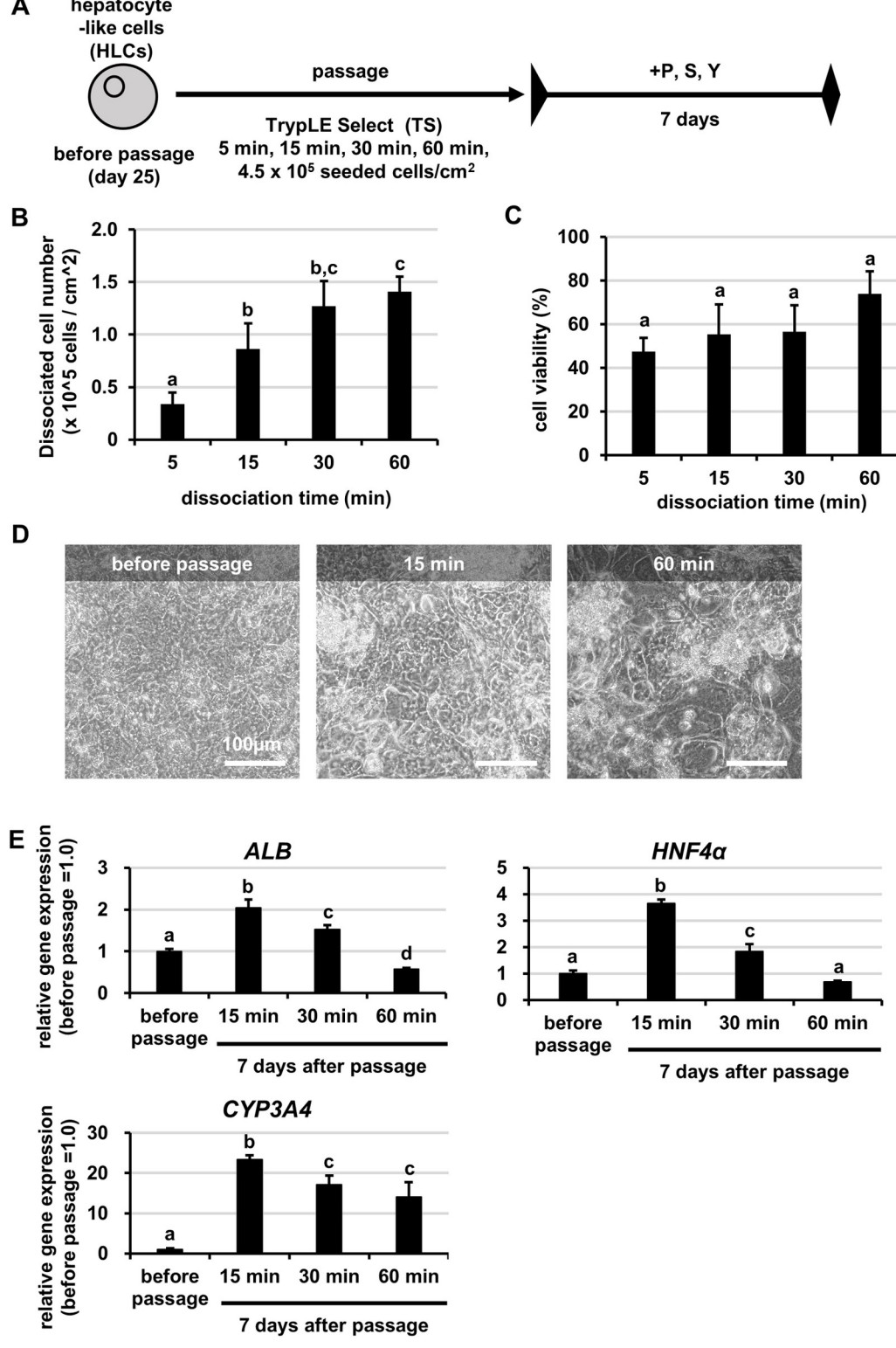

**Fig 3. Optimum cell-dissociation time for passage of HLCs.** Human iPS cells (Tic) were differentiated into hepatocyte-like cells (HLCs) as described in the Materials and Methods section. (A) The schematic overview shows the protocol for passage of HLCs. HLCs were passaged under four different cell dissociation times. (B) The number of dissociated cells per cell culture area was counted at different dissociation times. (C) Cell viability was calculated under different dissociation times. (D) Phase-contrast micrographs of HLCs before passage and HLCs cultured for 7 days after passage under different

dissociation times. (E) The gene expression levels of hepatocyte markers (*ALB*, *HNF4α*, *CYP3A4*) were examined in HLCs before passage and HLCs cultured for 7 days after passage under different dissociation times by real-time RT-PCR. The gene expression levels in HLCs before passage were taken as 1.0. Data represent the means ± SD (n = 3). Statistical significance was evaluated by one-way ANOVA followed by Tukey's post-hoc tests to compare all groups. Groups that do not share the same letter are significantly different from each other (p<.05).

### Cryopreservation of HLCs

Finally, we attempted cryopreservation of HLCs by applying the present passage method. Following the protocol shown in **Fig 6A**, HLCs were dissociated by the present passage method and cryopreserved for 7 days in three different freezing solutions (CELLBANKER 1, CB1; STEM-CELLBANKER, SCB; CELLBANKER 2, CB2). The cryopreserved HLCs were thawed and seeded. As a control, HLCs were passaged, re-seeded, and cultured without cryopreservation. After 7 days in culture, their cell morphology and expression levels of hepatocyte marker genes were evaluated. Cell morphology in all the cryopreserved groups was comparable to that of the control without cryopreservation and before passage (**Fig 6B**). Gene expression levels in all cryopreserved groups were maintained at levels comparable to or higher than those before passage, especially gene expression levels of *ALB*, *CYP3A4*, *CYP3A5*, *CYP3A7*, *CYP2B6*, *BSEP* and *MDR1* were particularly elevated compared to those before passage (**Figs 6C and S8**). CB1 was found to be the optimal freezing solution for cryopreservation of HLCs in our experiments. We compared albumin secretion, urea secretion, and CYP3A4 activity in HLCs after passage or cryopreservation in CB1 with those in HLCs before passage. The results showed that HLCs had these functions after passage or cryopreservation comparable to or better than those in HLCs before passage (Fig 7). In particular, albumin secretion increased significantly in HLCs after passage, and this trend was maintained even after cryopreservation. Thus, it was suggested that the application of the present passage method improves hepatic functions in HLCs even after cryopreservation.

## Discussion

Passage and culture of HLCs without loss of hepatic functions and improvement of hepatic functions are the most important factors in terms of fostering the wide use of HLCs in both pharmaceutical studies and regenerative medicine. In this study, we have developed a passage method that prevents the loss of HLC functions due to passage. In addition, the present passage method improved hepatic gene expressions and CYP3A4 activity in HLCs. This indicates that the post-passage HLCs would be suitable for use in the evaluation of drug toxicity in drug discovery. Finally, we showed that the HLCs also maintained their hepatic functions after cryopreservation. Therefore, by applying this technology, it will be possible to provide ready-to-use availability of cryopreserved HLCs for drug discovery research.

In this study, we found that EMT was one of the possible causes of the functional decline of HLCs after passage. EMT inhibitors were effective in suppressing EMT and maintaining the hepatic functions after passage. Further, this study found that optimization of cell dissociation time improved hepatic gene expressions in HLCs after passage compared to those before passage. The suppression of EMT in the passaged HLCs by EMT inhibitors appeared to play a major role in the improved hepatic gene expressions and CYP3A4 activity, but other effects might also have been involved. One possible reason for the improved hepatic gene expressions and CYP3A4 activity in HLCs after passage could be the enrichment of HLCs having higher hepatic gene expressions by passage manipulation. To determine whether the HLCs having higher hepatic gene expressions were sorted by passages, we compared the gene expression levels of hepatic markers between the HLCs just after the passage manipulation (0 hr after

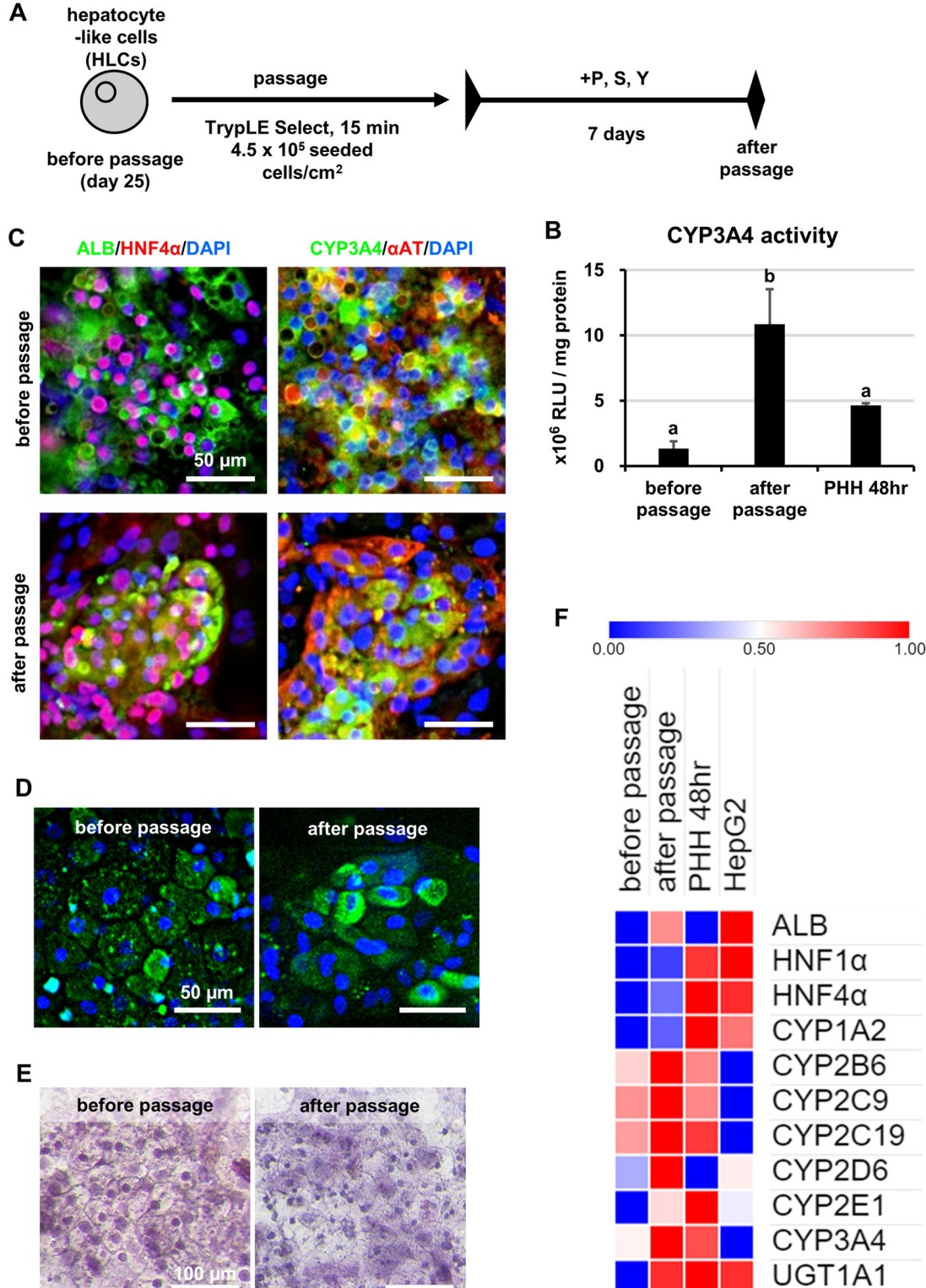

**Fig 4. Characteristics of HLCs passaged under optimized passage conditions.** Human iPS cells (Tic) were differentiated into hepatocyte-like cells (HLCs) as described in the Materials and Methods section. (A) The schematic overview shows the protocol for passage of HLCs. (B) The CYP3A4 activity was examined in HLCs before passage, HLCs cultured for 7 days after passage and PHHs cultured for 48 hours (PHH 48hr). Statistical significance was evaluated by one-way ANOVA followed by Tukey's post-hoc tests to compare all groups. Groups that do not share the same letter are significantly different from each other ($p < .05$). (C)

The expression of the hepatocyte markers (ALB, HNF4α, CYP3A4, αAT) in HLCs before and after passage was examined by immunohistochemistry. (D) LDL uptake was examined in HLCs before and after passage by Alexa-488-labeled LDL. (E) Cytoplasmic accumulation of glycogen was determined in HLCs before and after passage by PAS staining. (F) Comparative analysis of the gene expression levels of major hepatocyte markers was performed between HLCs before passage, HLCs after passage, PHHs cultured for 48 hours (PHH 48hr) and HepG2 cells. The gene expression levels of hepatocyte markers were examined by real-time RT-PCR. Heatmap was generated using Morpheus (https://software.broadinstitute.org/morpheus) by comparing and normalizing the gene expressions (ΔΔCt) between HLCs before passage, HLCs after passage, PHHs cultured for 48 hours (PHH 48hr) and HepG2 cells.

passage) and those cultured for 7 days after passage (S9 Fig). The results showed that the gene expression levels of hepatocyte markers in HLCs immediately after passage (0 hr after passage) were not significantly different compared to those before passage, and that they were enhanced by 7 days of culture. Thus, HLCs having higher hepatic gene expressions were not likely to be selectively recovered by the present passage method. Another possible reason is off-target effects of the EMT inhibitors. Further examination is necessary to elucidate the mechanism underlying improved hepatic gene expressions in HLCs after passage.

Based on a single-cell RNA sequencing analysis, Chembazhi *et al.* reported the presence of a population of hepatocytes in a hypermetabolic state during liver regeneration in hepatecto-mized mice [21]. It is thought that the hepatocytes responsible for proliferation during liver regeneration have impaired hepatic functions, and other hepatocytes compensate for the deficiency by activating their metabolic functions. If a similar phenomenon occurs in the human liver, it may explain the improved hepatic gene expressions and CYP3A4 activity in HLCs after passage. Oliva *et al.* reported that when 3D spheroids were formed from PHHs, the gene expression variation recapitulated the liver regeneration process, and the gene expression levels of hepatic progenitor cell markers such as *EpCAM* increased during the early stages of 3D spheroid formation [22]. In this study, we also observed increased gene expression levels of several hepatic progenitor cell markers, including *EpCAM*, in HLCs after passage (S10 Fig). Thus, it was suggested that HLCs after passage may mimic liver regeneration. In the future, comprehensive analysis of gene expression changes in HLCs before and after passage by RNA-seq may help to identify new factors involved in the improved hepatic gene expressions in HLCs after passage.

Recently, microphysiological systems (MPS) have been attracting attention because they can better reproduce the physiology of the human liver [23]. When applying HLCs to MPS, it is essential to passage, seed and culture HLCs into the MPS device. The present passage method would thus be useful for the application of HLCs to drug discovery research using MPS.

In summary, we have developed a passage method that suppresses the functional decline of HLCs after passage. Using the present passage method, we have shown that the hepatic gene expressions and CYP3A4 activity in HLCs improve even after passage and that HLCs can be cryopreserved. We believe that the passage method developed in this study will accelerate drug discovery research using HLCs.

## Materials and methods

### Human iPS cells culture

The human iPS cell lines Tic (obtained from the JCRB Cell Bank, JCRB Number: JCRB1331) and DOO-iPS [4] cells were maintained on 1 μg/cm$^2$ recombinant human laminin 511 E8 fragments (iMatrix-511, Nippi, Tokyo, Japan) with StemFit AK02N medium (Ajinomoto). To passage human iPS cells, near-confluent human iPS cell colonies were treated with TrypLE Select Enzyme (Thermo Fisher Scientific) for 5 minutes at 37˚C. After centrifugation, human

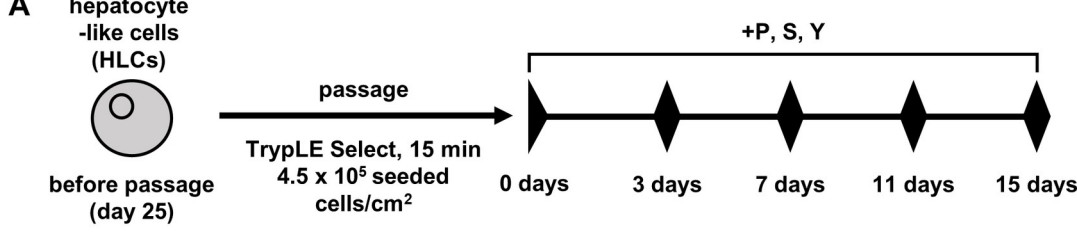

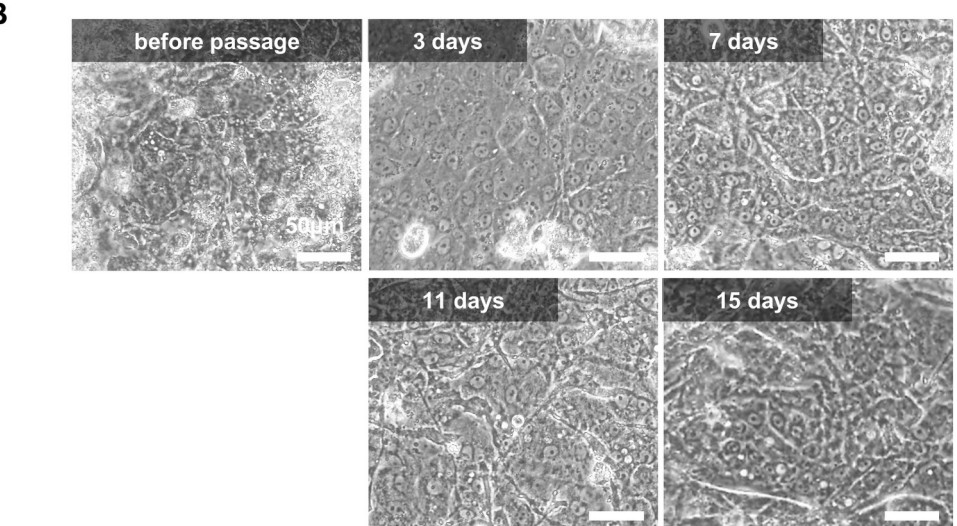

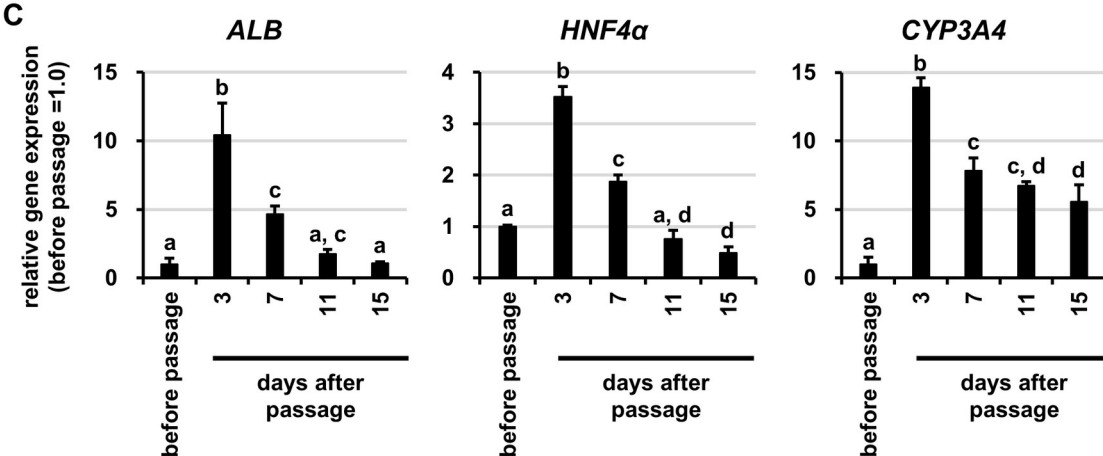

**Fig 5. Extended culture of HLCs after passage.** Human iPS cells (Tic) were differentiated into hepatocyte-like cells (HLCs) as described in the Materials and Methods section. (A) The schematic overview shows the protocol for passage of HLCs. (B) Phase-contrast micrographs of HLCs before passage and HLCs cultured for 3 to 15 days after passage. (C) The gene expression levels of hepatocyte markers (*ALB*, *HNF4α*, *CYP3A4*) were examined in HLCs before passage and HLCs cultured for 3 to 15 days after passage by real-time RT-PCR. The gene expression levels in HLCs before passage were taken as 1.0. Data represent the means ± SD (n = 3). Statistical significance was evaluated by one-way ANOVA followed by Tukey's post-hoc tests to compare all groups. Groups that do not share the same letter are significantly different from each other (p<.05).

iPS cells were seeded at an appropriate cell density ($5 \times 10^4$ cells/cm$^2$) onto iMatrix-511 and were then subcultured every 6 days.

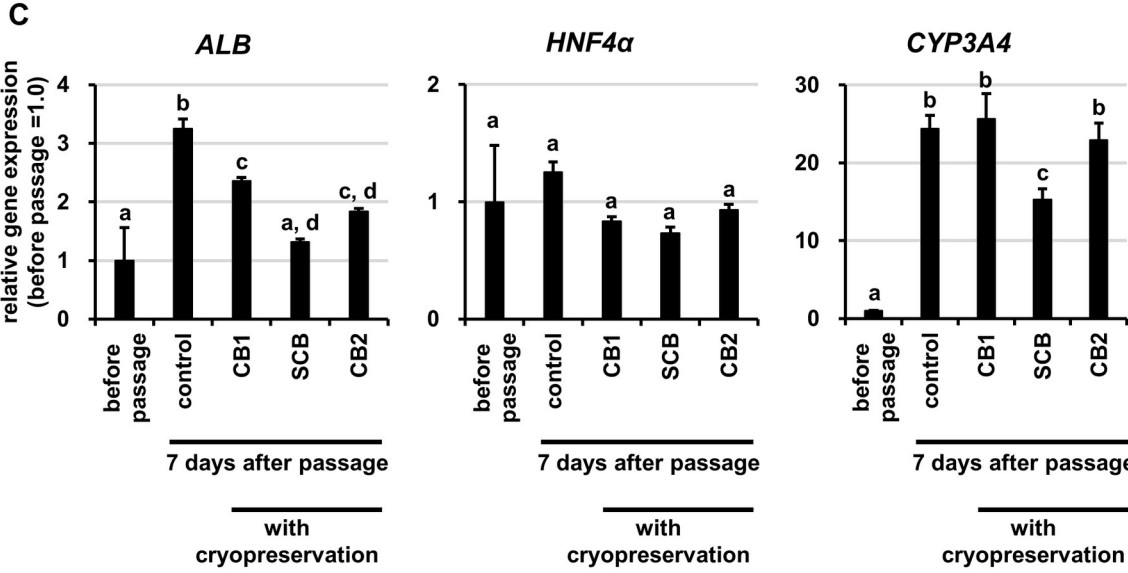

**Fig 6. Cryopreservation of HLCs.** Hepatocyte-like cells (HLCs) were differentiated from human iPS cells (Tic) and cryopreserved as described in the Materials and Methods section. (A) The schematic overview shows the protocol for cryopreservation and seed of HLCs. To freeze HLCs, program Deep Freezer (PDF300, STREX) and freezing solutions (CELLBANKER 1, CB1; STEM-CELLBANKER, SCB; CELLBANKER 2, CB2) were used. (B) Phase-contrast micrographs of HLCs before passage, HLCs cultured for 7 days after passage (control) and cultured for 7 days after cryopreservation. (C) The gene expression levels of hepatocyte markers (*ALB*, *HNF4α*, *CYP3A4*) were examined in HLCs before passage, HLCs cultured for 7 days after passage (control) and cultured for 7 days after cryopreservation

by real-time RT-PCR. The gene expression levels in HLCs before passage were taken as 1.0. Data represent the means ± SD (n = 3). Statistical significance was evaluated by one-way ANOVA followed by Tukey's post-hoc tests to compare all groups. Groups that do not share the same letter are significantly different from each other (p<.05).

### Hepatic differentiation

Before the initiation of hepatic differentiation, human iPS cells were dissociated into single cells by using TrypLE Select Enzyme and plated onto Matrigel-coated dishes. The cells were then cultured in StemFit AK02N medium for 24 hours. The differentiation protocol for the induction of definitive endoderm cells, hepatoblast-like cells, and HLCs was based on our previous reports [4] with some modifications. Briefly, in the definitive endoderm differentiation, human iPS cells were cultured for 4 days in RPMI1640 medium (Sigma-Aldrich), which contained 100 ng/mL Activin A (R&D Systems), 2× GlutaMAX (Thermo Fisher Scientific), and 0.5× B27 Supplement Minus Vitamin A (Thermo Fisher Scientific). For the induction of hepatoblast-like cells, the definitive endoderm cells were cultured for 5 days in RPMI1640 medium containing 20 ng/mL BMP4 (R&D Systems), 20 ng/mL fibroblast growth factor 4 (FGF4; R&D Systems), 2× GlutaMAX, and 0.5× B27 Supplement Minus Vitamin A. To perform hepatic differentiation, the hepatoblast-like cells were cultured for 5 days in RPMI1640 medium containing 20 ng/mL hepatocyte growth factor, 2× GlutaMAX, and 0.5× B27 Supplement Minus Vitamin A. Finally, the cells were cultured for 11 days in hepatocyte culture medium (HCM, Lonza) without EGF but with 20 ng/mL oncostatin M (OsM; R&D Systems) and 3× GlutaMAX.

### Passage of human iPS cell-HLCs

HLCs were dissociated into single cells using cell dissociation enzymes and mechanical disruption with a pipette, and then passed through a 200 μm cell strainer (pluriSelect Life Science UG & Co. KG) and a 70 μm cell strainer (Falcon). As cell dissociation enzymes, TrypLE Select Enzyme (Thermo Fisher Scientific), Dispase & Collagenase, or Accutase (Millipore) was used. Dispase & Collagenase was a mixture of 1 mg/mL dispase (Roche) and 1 mg/mL collagenase (SERVA Electrophoresis GmbH). The cells were seeded at $4.5 \times 10^5$ cells/cm$^2$ in Matrigel-coated 48-well plates and cultured in HCM containing 10% fetal bovine serum (FBS; Gibco), 20 ng/mL OsM and 3 × GlutaMAX. Twenty-four hours later, the medium was replaced with HCM medium containing 20 ng/mL OsM and 3 × GlutaMAX. Thereafter, the medium was changed every 2 days. After passage, PD0325901 (0.5 μM, Wako), SB43154 (2 μM, Wako), Y-27632 (10 μM, Wako), and DMSO (final concentration 0.2%, Wako) were added to the medium as needed to culture HLCs.

### Real-time RT-PCR

Total RNA was extracted from each cell population using ISOGENE (NIPPON GENE). cDNA was synthesized from 500 ng of each Total RNA by reverse transcription reaction using Superscript VILO cDNA synthesis kit (Thermo Fisher Scientific). Target mRNA expression levels were quantified relatively using the 2-ΔΔCT method. Glyceraldehyde 3-phosphate dehydrogenase (GAPDH) was used as an internal standard gene. Primer sequences used for quantitative RT-PCR (Table 1) were obtained from PrimerBank (https://pga.mgh.harvard.edu/primerbank/).

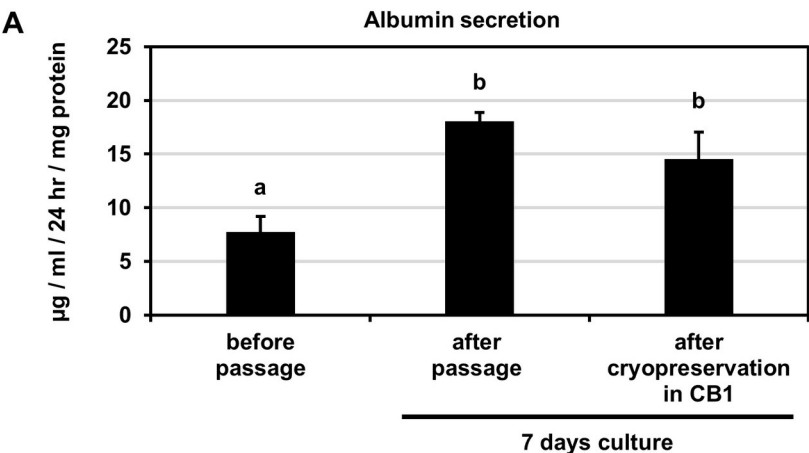

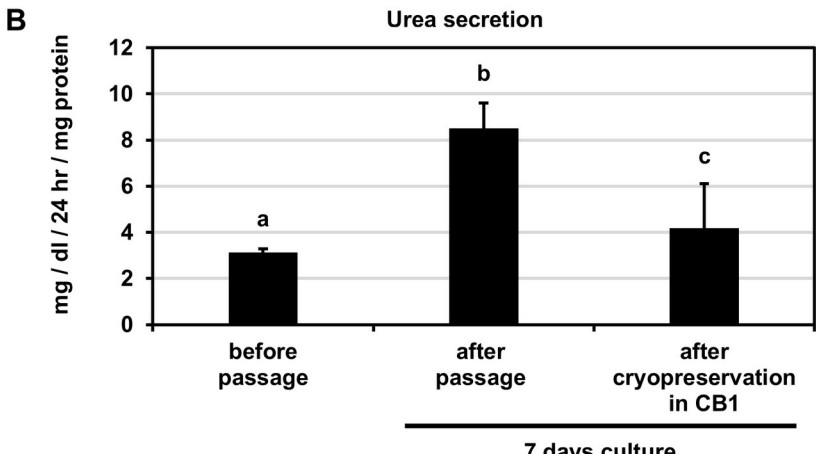

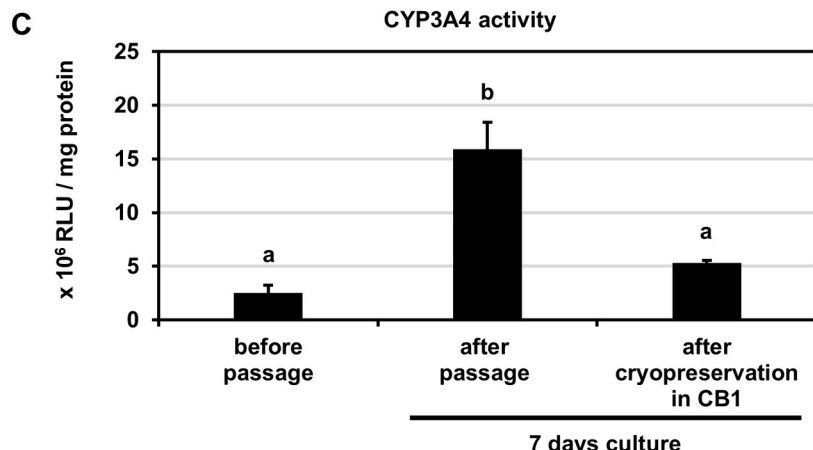

**Fig 7. Albumin secretion, urea secretion, and CYP3A4 activity in HLCs after passage and cryopreservation.**
Hepatocyte-like cells (HLCs) were differentiated from human iPS cells (Tic) and cryopreserved as described in the Materials and Methods section. The albumin secretion (A), urea secretion (B), and CYP3A4 activity (C) were examined in HLCs before passage, HLCs cultured for 7 days, HLCs cultured for 7 days after cryopreservation in CB1. Data represent the means ± SD (n = 3). Statistical significance was evaluated by one-way ANOVA followed by Tukey's post-hoc tests to compare all groups. Groups that do not share the same letter are significantly different from each other (p<.05).

**Table 1. The primers used for real-time RT-PCR.**

| Gene Symbol | for real-time RT-PCR |
| --- | --- |
|  | Primers (forward/reverse; 5' to 3') |
| GAPDH | GGTGGTCTCCTCTGACTTCAACA/GTGGTCGTTGAGGGCAATG |
| ALB | TGCAACTCTTCGTGAAACCTATG/ACATCAACCTCTGGTCTCACC |
| HNF4α | CGTCATCGTTGCCAACACAAT/GGGCCACTCACACATCTGTC |
| CYP3A4 | AAGTCGCCTCGAAGATACACA/AAGGAGAGAACACTGCTCGTG |
| SNAI1 | AGGTTGGAGCGGTCAGC/CCTTCTCTAGGCCCTGGCT |
| Fibronectin | AGGAAGCCGAGGTTTTAACTG/AGGACGCTCATAAGTGTCACC |
| E-cadherin | CGAGAGCTACACGTTCACGG/GGGTGTCGAGGGAAAAATAGG |
| EpCAM | AATCGTCAATGCCAGTGTACTT/TCTCATCGCAGTCAGGATCATAA |
| PROX1 | TTGACATTGGAGTGAAAAGGACG/TGCTCAGAACCTTGGGGATTC |
| CK7 | AGACGGAGTTGACAGAGCTG/GGATGGCCCGGTTCATCTC |
| αAT | ATGCTGCCCAGAAGACAGATA/CTGAAGGCGAACTCAGCCA |
| HNF1α | AACACCTCAACAAGGGCACTC/CCCCACTTGAAACGGTTCCT |
| CYP3A5 | CGGCATCATAGGTAGGTGGT/TATGAACTGGCCACTCACCC |
| CYP3A7 | AAGGTCGCCTCAAAGAGACA/TGCACTTTCTGCTGGACATC |
| CYP2B6 | GTCCCAGGTGTACCGTGAAG/CCCTTTTGGGAAACCTTCTG |
| UGT1A1 | TGACGCCTCGTTGTACATCAG/CCTCCCTTTGGAATGGCAC |
| TAT | TACAGACCCTGAAGTTACCCAG/TAAGAAGCAATCTCCTCCCGA |
| BSEP | TGATCCTGATCAAGGGAAGG/TGGTTCCTGGGAAACAATTC |
| OATP2 | TAAAGCTGAGTGACAGAGCTGC/AAACAGCAGAGGCACAACCT |
| MDR1 | GCCAAAGCCAAAATATCAGC/TTCCAATGTGTTCGGCATTA |
| BCRP | TGCAACATGTACTGGCGAAGA/TCTTCCACAAGCCCCAGG |
| MRP2 | TGAGCAAGTTTGAAACGCACAT/AGCTCTTCTCCTGCCGTCTCT |
| CAR | TAATGCGCTGACTTGTGAGG/TCATGCCAGCATCTAAGCAC |

## Immunofluorescence

To perform the immunofluorescence, cells were washed twice with PBS and treated with 4% paraformaldehyde (FUJIFILM Wako Pure Chemical) for 10 min at room temperature. Cells were blocked with PBS containing 2% BSA (Sigma-Aldrich) and 0.2% Triton X-100 (Sigma-Aldrich) for 45 min. Cells were reacted with primary antibody overnight at 4°C, followed by secondary antibody (Thermo Fisher Scientific) labeled with Alexa Fluor 488 or Alexa Fluor 594 for 1 hour at room temperature. Nuclear staining was then performed using 4',6-diami-dino-2-phenylindole (DAPI) (Thermo Fisher Scientific), and cells were observed under a fluorescence microscope (BIOREVO BZ-9000, Keyence). Antibodies used are listed in Table 2.

**Table 2. The antibodies used for immunocytochemistry.**

| Antigen | Type | Company | Catalog number | Dilution factor |
| --- | --- | --- | --- | --- |
| ALB | goat | Bethyl Laboratories | A80-129A | 1:200 |
| CYP3A4 | goat | Santa Cruz Biotechnology | sc-27639 | 1:50 |
| HNF4α | rabbit | Santa Cruz Biotechnology | sc-374229 | 1:200 |
| αAT | rabbit | Dako | A0012 | 1:200 |
| Alexa Fluor 488 anti-goat IgG | donkey | Thermo Fisher Scientific | A11055 | 1:1000 |
| Alexa Fluor 594 anti-rabbit IgG | donkey | Thermo Fisher Scientific | A21207 | 1:1000 |

### FACS analysis

Single-cell suspensions were fixed with 4% PFA at 4°C for 10 min, and then incubated with the primary antibody, followed by the secondary antibody. Analysis was performed on a MACSQuant Analyzer (Miltenyi Biotec) and FlowJo software (FlowJo LLC, http://www. flowjo.com/). Antibodies used are listed in Table 2.

### CYP3A4 activity measurement

CYP3A4 activity was measured using P450-Glo™ CYP3A4 Assay Kits (Promega). Luciferin-IPA was used as a substrate for CYP3A4, and fluorescence intensity was measured with a luminometer (Lumat LB 9507, Berthold). CYP3A4 activity was corrected by the amount of protein per well.

### PAS staining

Cells were washed with PBS and stained using PERIODIC ACID-SCHIFF (PAS) STAINING SYSTEM (Sigma-Aldrich) according to the instruction manual.

### LDL uptake capacity

Cells were cultured in medium supplemented with Alexa-488-labeled LDL (Invitrogen) for 1 hour and washed twice with PBS. Then, nuclear staining was performed using 4',6-diamidino-2-phenylindole (DAPI) (Thermo Fisher Scientific), followed by staining with 4% paraformaldehyde (FUJIFILM Wako Pure Chemical). After fixation with a fluorescent microscope (BIOREVO BZ-9000, Keyence), the samples were observed under a fluorescence microscope (BIOREVO BZ-9000, Keyence).

### Cryopreservation of human iPS cell-HLCs

Single-celled HLCs were suspended in freezing solution and frozen to -80°C at -1°C per minute in the program Deep Freezer (PDF300, STREX). HLCs were then cryopreserved in a -150°C freezer for 7 days. As freezing solutions, CELLBANKER 1 (ZENOAQ), STEM-CELL-BANKER (DMSO Free GMP grade, ZENOAQ), CELLBANKER 2 (ZENOAQ) were used.

### ALB secretion

The culture supernatants, which were incubated for 24 hr after fresh medium was added, were collected and analyzed by Enzyme-Linked Immuno Sorbent Assay (ELISA) to determine their levels of ALB secretion. A Human Albumin ELISA Quantitation Set was purchased from Bethyl Laboratories. ELISA was performed according to the manufacturer's instructions. The amount of ALB secretion was calculated according to each standard followed by normalization to the protein content per well. Protein content was measured using Pierce BCA Protein Assay Kit according to the manufacturer's instructions.

### Urea secretion

The culture supernatants, which were incubated for 24 hr after fresh medium was added, were collected and analyzed for the amount of urea production. Urea measurement kits were purchased from BioAssay Systems. The experiment was performed according to the manufacturer's instructions. The amount of urea secretion was calculated according to each standard followed by normalization to the protein content per well. Protein content was measured using Pierce BCA Protein Assay Kit according to the manufacturer's instructions.

**Table 3. The background of primary human hepatocyte.**

| lot | sex | age | race | cause of death |
|-----|-----|-----|------|----------------|
| FCL | female | 10 month | Hispanic or Latino Origin | anoxia/drowning |
| OHO | male | 28 years | Caucasian | Head Trauma |
| YEM | female | 46 years | Caucasian | ICH-Stroke |

### Primary human hepatocytes

Three lots of cryopreserved human hepatocytes (lots: OHO, FCL, YEM: Veritas) were used (Table 3). The data for the primary human hepatocytes (PHHs) are the average values of the three lots. The vials of hepatocytes were rapidly thawed in a shaking water bath at 37°C; the contents of each vial were emptied into prewarmed Cryopreserved Hepatocyte Recovery Medium (CHRM, Thermo Fisher Scientific) and the suspension was centrifuged at 900rpm for 10min at room temperature. The hepatocytes were seeded at $1.25 \times 10^5$ cells/cm$^2$ in HCM containing 10% fetal calf serum (FCS) onto type I collagen (Nitta Gelatin)-coated 12-well plates. The human hepatocytes cultured for 48hr after plating were used in the experiments.

### Statistical analysis

All results are based on three biological replicates. Statistical analysis was performed using the unpaired two-tailed Student's t-test or one-way ANOVA followed by Tukey's post-hoc tests. A value of $p < 0.05$ was considered statistically significant.

### Supporting information

**S1 Fig. Effect of passage on gene expression levels of cytochrome P450 2C9 and 2C19.**
(TIF)

**S2 Fig. Effects of the EMT inhibitors on HLCs.**
(TIF)

**S3 Fig. Effect of passage under different cell dissociation conditions on gene expression levels of CYP2C9 and 2C19.**
(TIF)

**S4 Fig. Effects of the optimum passage procedures on the hepatic gene expressions in different iPS cell line-derived HLCs.**
(TIF)

**S5 Fig. Effects of cell dissociation enzymes on HLCs.**
(TIF)

**S6 Fig. Comparison of the percentage of albumin- and CYP3A4-positive HLCs before and after passage.**
(TIF)

**S7 Fig. Effect of HLCs passage on gene expression levels of various hepatocyte markers.**
(TIF)

**S8 Fig. Effect of HLCs cryopreservation on gene expression levels of other hepatocyte markers.**
(TIF)

**S9 Fig. Culture of HLCs after passage was required for the enhanced hepatic functions.**
(TIF)

**S10 Fig. Transient dedifferentiation in HLCs after passage.**
(TIF)

## Acknowledgments

We thank Dr. Kazuo Takayama for his excellent advice.

## Author Contributions

**Conceptualization:** Hiroyuki Mizuguchi.

**Data curation:** Jumpei Inui, Yukiko Ueyama-Toba.

**Formal analysis:** Jumpei Inui, Yukiko Ueyama-Toba, Seiji Mitani.

**Funding acquisition:** Yukiko Ueyama-Toba, Hiroyuki Mizuguchi.

**Investigation:** Jumpei Inui, Yukiko Ueyama-Toba, Seiji Mitani.

**Methodology:** Jumpei Inui, Yukiko Ueyama-Toba, Seiji Mitani.

**Project administration:** Hiroyuki Mizuguchi.

**Resources:** Hiroyuki Mizuguchi.

**Supervision:** Hiroyuki Mizuguchi.

**Validation:** Jumpei Inui, Yukiko Ueyama-Toba, Seiji Mitani.

**Visualization:** Jumpei Inui, Yukiko Ueyama-Toba.

**Writing – original draft:** Jumpei Inui, Hiroyuki Mizuguchi.

**Writing – review & editing:** Jumpei Inui, Yukiko Ueyama-Toba, Seiji Mitani, Hiroyuki Mizuguchi.

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
