## [Decision Letter · Decision Letter 0]

19 Dec 2022

PONE-D-22-31940Development of a method of passaging and freezing human iPS cell-derived hepatocytes to improve their functions.PLOS ONE

Dear Dr. Mizuguchi,

Thank you for submitting your manuscript to PLOS ONE. After careful consideration, we feel that it has merit but does not fully meet PLOS ONE’s publication criteria as it currently stands. Therefore, we invite you to submit a revised version of the manuscript that addresses the points raised during the review process.

Changes required for acceptance:1. Functional assays such as albumin ELISA and urea assay should be performed.2. Quantification of immunostaining results should be presented (e.g. Fig. 4C, D, E). Recommended changes:1. The quality of figures should be improved.2. Alternative cell dissociation methods should be explored.3. Other recommended changes as suggested by the reviewers. Please submit your revised manuscript by Feb 02 2023 11:59PM. If you will need more time than this to complete your revisions, please reply to this message or contact the journal office at plosone@plos.org. Please include the following items when submitting your revised manuscript:A rebuttal letter that responds to each point raised by the academic editor and reviewer(s). You should upload this letter as a separate file labeled 'Response to Reviewers'.A marked-up copy of your manuscript that highlights changes made to the original version. You should upload this as a separate file labeled 'Revised Manuscript with Track Changes'.An unmarked version of your revised paper without tracked changes. You should upload this as a separate file labeled 'Manuscript'.

We look forward to receiving your revised manuscript.

Kind regards,

Hon Fai Chan, PhD

Academic Editor

PLOS ONE

Reviewers' comments:

Reviewer's Responses to Questions

**Comments to the Author**

1. Is the manuscript technically sound, and do the data support the conclusions?

Reviewer #1: Partly

Reviewer #2: Yes

2. Has the statistical analysis been performed appropriately and rigorously? 

Reviewer #1: N/A

Reviewer #2: Yes

3. Have the authors made all data underlying the findings in their manuscript fully available?

Reviewer #1: Yes

Reviewer #2: Yes

4. Is the manuscript presented in an intelligible fashion and written in standard English?

Reviewer #1: Yes

Reviewer #2: Yes

5. Review Comments to the Author

Reviewer #1: The passaging and freezing of human iPS cell-derived hepatocytes is an important issue in this study. The authors describe methods for dissociating, cryopreserving and reseeding HLCs and conclude that hepatic function persists. However, the main problem with this paper is that the data lack quantification and controls to fully support the conclusions of the paper.

1) Fig4C lacks quantification and does not compare the ratio of ALB HNF4A CYP3A4 AAT before and after passaging. neither does Fig4D and E.

2) Fig4B suggests using ES as a negative control and Fresh PHH as a positive control.

3) The full text should be quantified using the ALB Secretion ELISA assay.

4) Fig6, Comparison of cell activity after recovery with different cryopreservation solutions. In addition to qPCR, more functional quantitative data should be used for comparison.

Reviewer #2: The paper of Mizuguchi et al. describes an improved method for passaging and freezing hepatocyte-like cells derived from iPSC. The topic is interesting since PSC-derived models present many advantages such their availability and may offer a stable source of hepatocytes and could be maintained for longer periods, although the present protocols still present many limitations like immature phenotype or difficulties for passaging and cryopreserving them. The paper is well designed and written but it has limitations that make it not suitable for publication in its present. A detailed list if the points needing revision is provided below:

1. The quality of the figures should be deeply improved. The schemes used are so simplistic and could be also improved and make them more appealing.

2. The authors just analyzed the use of TriplE for passaging the cells although many authors have used other systems such as collagenase, dispase or cell dissociation buffer for cell passaging. The authors should explore the use of other methods that just require 5-10 minutes.

3. A more detailed transcriptomics analysis (at least 10 genes) would improve the quality of the manuscript.

4. Functionality of the cells in terms of albumin production and secretion and ureogenesis capability should be also included since these are specific hepatic functions that are normally analysed in HLC.

6. PLOS authors have the option to publish the peer review history of their article (what does this mean?). If published, this will include your full peer review and any attached files.

Reviewer #1: No

Reviewer #2: No

---

## [Author Response · Author response to Decision Letter 0]

18 Apr 2023

Dear Editor of PLoS ONE

Thank you for your effort in reviewing our manuscript entitled “Development of a method of passaging and freezing human iPS cell-derived hepatocytes to improve their functions.” by J. Inui et al. We thoroughly revised the manuscript in keeping with editor’s and reviewers’ suggestions. The sentences we revised to the manuscript are shown in red. Point-by-point responses about editor’s and reviewers’ suggestions are given below. 

Changes required for acceptance:

1. Functional assays such as albumin ELISA and urea assay should be performed.

Response: 

We compared albumin secretion, urea secretion, and CYP3A4 activity in HLCs after passage or cryopreservation in CB1 with those in HLCs before passage. The results showed that HLCs had these functions after passage or cryopreservation comparable to or better than those in HLCs before passage.

Please see lines 216-221 and Fig.7.

2. Quantification of immunostaining results should be presented (e.g. Fig. 4C, D, E).

Response: 

We found it difficult to quantify the immunofluorescent staining images. Therefore, we instead performed FACS analysis about the expression of the most important hepatocyte markers, ALB and CYP3A4. The results showed that the percentage of ALB- or CYP3A4-positive cells increased in the HLCs after passage compared to before passage. Please see lines 170–174 and Fig.S6. 

In the results in Fig. 4D and E, the analyses were performed to confirm that HLCs after passage still had the capacity for glycogen storage and LDL uptake, and did not indicate that these functions were higher than before passage.

Recommended changes:

1. The quality of figures should be improved.

Response:

We corrected the data with increased resolution.

2. Alternative cell dissociation methods should be explored.

Response:

We showed the results using other dissociation enzymes [dispase & collagenase (DC), TrypLE select (TS) and accutase (Ac)] in Fig. S5, and decided to use TrypLE select, which showed a more favorable trend, although not significantly different. We tried to dissociate HLCs with 5 minutes treatment of TrypLE select, but the cells were not recovered efficiently enough to perform the experiments, as shown in Fig. 3B.

3. Other recommended changes as suggested by the reviewers.

Response:

We thoroughly revised the manuscript and figures in keeping with each reviewer's suggestions. Point-by-point responses about reviewers’ suggestions are given below.

Reviewer #1:

1. Fig4C lacks quantification and does not compare the ratio of ALB HNF4A CYP3A4 AAT before and after passaging. neither does Fig4D and E.

Response: 

　As the reviewer suggested, we performed FACS analysis to determine the percentage of ALB- or CYP3A4-positive cells before and after passage. The results showed that the percentage of ALB- or CYP3A4-positive cells increased in the HLCs after passage compared to before passage. Please see lines 170–174 and Fig.S6. 

In the results in Fig. 4D and E, the analyses were performed to confirm that HLCs after passage still had the capacity for glycogen storage and LDL uptake, and did not indicate that these functions were higher than those before passage.

2. Fig4B suggests using ES as a negative control and Fresh PHH as a positive control.

Response:

 Since HLCs are derived from human iPS cells, we measured CYP3A4 activity in human iPS cells as a negative control and found no activity (data not shown). Most pharmaceutical companies perform drug toxicity studies with PHHs cultured for 48 to 72 hours. Therefore, in this study, we used PHHs cultured for 48 hours as a control to demonstrate the usefulness of HLCs after passage for drug toxicity studies.

3. The full text should be quantified using the ALB Secretion ELISA assay.

Response: 

　As the reviewer suggested, we compared the levels of albumin secretion in HLCs after passage or cryopreservation with those in HLCs before passage. The results showed that the levels of albumin secretion increased significantly in HLCs after passage, and this trend was maintained even after cryopreservation. Please see lines 216-221 and Fig.7.

4. Fig6, Comparison of cell activity after recovery with different cryopreservation solutions. In addition to qPCR, more functional quantitative data should be used for comparison.

Response: 

　As the reviewer suggested, we compared albumin secretion, urea secretion, and CYP3A4 activity in HLCs after passage or cryopreservation in CB1 with those in HLCs before passage. The results showed that HLCs after cryopreservation had these functions comparable to or better than the HLCs before passage, although urea secretion and CYP3A4 activity in the HLCs after cryopreservation were lower than those in the HLCs after passage. Please see lines 216-221 and Fig.7.

 

Reviewer #2: 

1. The quality of the figures should be deeply improved. The schemes used are so simplistic and could be also improved and make them more appealing.

Response:

We corrected the data with increased resolution. 　

2. The authors just analyzed the use of TriplE for passaging the cells although many authors have used other systems such as collagenase, dispase or cell dissociation buffer for cell passaging. The authors should explore the use of other methods that just require 5-10 minutes.

Response:

We showed the results using other dissociation enzymes [dispase & collagenase (DC), TrypLE select (TS) and accutase (Ac)] in Fig. S5, and decided to use TrypLE select, which showed a more favorable trend, although not significantly different. We tried to dissociate HLCs with 5 minutes treatment of TrypLE select, but the cells were not recovered efficiently enough to perform the experiments, as shown in Fig. 3B.

3. A more detailed transcriptomics analysis (at least 10 genes) would improve the quality of the manuscript.

Response: 

　As the reviewer suggested, we additionally examined the gene expression levels of more than 10 hepatocyte markers. Most genes were highly expressed in HLCs after passage or cryopreservation compared to those in HLCs before passage. Please see lines 185-187, 213-215, and Fig.S7 & S8.

4. Functionality of the cells in terms of albumin production and secretion and ureogenesis capability should be also included since these are specific hepatic functions that are normally analysed in HLC.

Response: 

　As the reviewer suggested, we compared albumin secretion and urea secretion in HLCs after passage or cryopreservation in CB1 with those in HLCs before passage. The results showed that HLCs had these functions after passage or cryopreservation comparable to or better than those in HLCs before passage, although urea secretion in the HLCs after cryopreservation was lower than that in the HLCs after passage. Please see lines 216-221 and Fig.7.

---

## [Decision Letter · Decision Letter 1]

2 May 2023

Development of a method of passaging and freezing human iPS cell-derived hepatocytes to improve their functions.

PONE-D-22-31940R1

Dear Dr. Mizuguchi,

We’re pleased to inform you that your manuscript has been judged scientifically suitable for publication and will be formally accepted for publication once it meets all outstanding technical requirements.

Kind regards,

Hon Fai Chan, PhD

Academic Editor

PLOS ONE

Additional Editor Comments (optional):

Reviewers' comments:

Reviewer's Responses to Questions

**Comments to the Author**

1. If the authors have adequately addressed your comments raised in a previous round of review and you feel that this manuscript is now acceptable for publication, you may indicate that here to bypass the “Comments to the Author” section, enter your conflict of interest statement in the “Confidential to Editor” section, and submit your "Accept" recommendation.

Reviewer #1: All comments have been addressed

Reviewer #2: All comments have been addressed

2. Is the manuscript technically sound, and do the data support the conclusions?

Reviewer #1: Partly

Reviewer #2: Yes

3. Has the statistical analysis been performed appropriately and rigorously? 

Reviewer #1: Yes

Reviewer #2: Yes

4. Have the authors made all data underlying the findings in their manuscript fully available?

Reviewer #1: Yes

Reviewer #2: No

5. Is the manuscript presented in an intelligible fashion and written in standard English?

Reviewer #1: Yes

Reviewer #2: Yes

6. Review Comments to the Author

Reviewer #1: The authors have supplied additional data according to the comments. The revised manuscript mostly meets the requirements. However, the quality of figures are still not good enough. It's difficult to identify the cell morphological differences.

Reviewer #2: (No Response)

7. PLOS authors have the option to publish the peer review history of their article (what does this mean?). If published, this will include your full peer review and any attached files.

Reviewer #1: No

Reviewer #2: No

---

## [Editor Report · Acceptance letter]

9 May 2023

PONE-D-22-31940R1 

Development of a method of passaging and freezing human iPS cell-derived hepatocytes to improve their functions. 

Dear Dr. Mizuguchi:

I'm pleased to inform you that your manuscript has been deemed suitable for publication in PLOS ONE. Congratulations! Your manuscript is now with our production department. 

Kind regards, 

on behalf of

Professor Hon Fai Chan 

Academic Editor

PLOS ONE